

# Judgement bias in goats (*Capra hircus*): investigating the effects of human grooming

Luigi Baciadonna, Christian Nawroth and Alan G. McElligott

Queen Mary University of London, Biological and Experimental Psychology, School of Biological and Chemical Sciences, London, UK

## ABSTRACT

Animal emotional states can be investigated by evaluating their impact on cognitive processes. In this study, we used a judgement bias paradigm to determine if short-term positive human-animal interaction (grooming) induced a positive affective state in goats. We tested two groups of goats and trained them to discriminate between a rewarded and a non-rewarded location over nine training days. During training, the experimental group ($n = 9$) was gently groomed by brushing their heads and backs for five min over 11 days (nine training days, plus two testing days, total time 55 min). During training, the control group ($n = 10$) did not experience any direct interaction with the experimenter, but was kept unconstrained next to him for the same period of time. After successful completion of the training, the responses (latency time) of the two groups to reach ambiguous locations situated between the two reference locations (i.e., rewarded/non-rewarded) were compared over two days of testing. There was not a positive bias effect after the animals had been groomed. In a second experiment, 10 goats were tested to investigate whether grooming induced changes in physiological activation (i.e., heart rate and heart rate variability). Heart rate increased when goats were groomed compared to the baseline condition, when the same goats did not receive any contact with the experimenter. Also, subjects did not move away from the experimenter, suggesting that the grooming was positively accepted. The very good care and the regular positive contacts that goats received from humans at the study site could potentially account for the results obtained. Good husbandry outcomes are influenced by animals' perception of the events and this is based on current circumstances, past experiences and individual variables. Taking into account animals' individual characteristics and identifying effective strategies to induce positive emotions could increase the understanding and reliability of using cognitive biases paradigms to investigate and promote animal welfare.

## INTRODUCTION

The assumption that animals are sentient and therefore able to experience emotions creates the new challenge of assessing their emotions and, when possible, to identify strategies to promote positive emotional experiences (*Panksepp, 2005*; *Burgdorf & Panksepp,*

Corresponding authors
Luigi Baciadonna,
luigi.baciadonna@qmul.ac.uk
Alan G. McElligott,
a.g.mcelligott@qmul.ac.uk

*2006*). The assessment of animal emotions is difficult, because animals cannot report emotional experiences through language (*Mendl et al., 2009*; *Mendl, Burman & Paul, 2010*; *Briefer, Tettamanti & McElligott, 2015*). The use of a multicomponent approach in which several parameters (e.g., behaviour and neurophysiology) are taken into account allows emotions to be assessed indirectly (*Briefer, Tettamanti & McElligott, 2015*; *Désiré, Boissy & Veissier, 2002*).

The use of the judgement bias in animals has been inspired by studies carried out in humans, because the ways that people perceive, interpret and judge information is influenced by their emotions and feelings (*Mendl et al., 2009*; *Boissy et al., 2007*). People with depression or anxiety are more likely to perceive social information as threatening and pay more attention to negative stimuli (*Nygren et al., 1996*). In animals, the impact of emotional states on behavioural expression has been identified through behavioural and physiological changes induced by specific environmental stimuli. For example, unfamiliar and unexpected objects generated a startle response in sheep (*Ovis aries*) (*Désiré, Boissy & Veissier, 2002*). Furthermore, when expectations about food were violated, lambs increased locomotor activity and there was a decrease in the parasympathetic influence on their cardiac activity (*Greiveldinger, Veissier & Boissy, 2011*).

Recently, the cognitive bias paradigm has been used to examine the interactions between emotional states and cognitive processes (e.g., attention, judgment and memory) in animals (*Mendl et al., 2009*; *Baciadonna & McElligott, 2015*; *Roelofs et al., 2016*). The assumption underlying this paradigm is that an experimentally induced alteration of an emotional state generates a behavioural response bias (e.g., judgement) that is linked with the emotional experience of the subject. Thus, the evaluation of ambiguous stimuli (i.e., novel stimuli introduced in between previously learned positive and negative cues) is affected by the emotional states experienced. There is strong evidence that the induction of negative emotional states generates a negative judgement of ambiguous stimuli (*Mendl et al., 2009*; *Baciadonna & McElligott, 2015*). For example, livestock exposed to long-term stressors (*Destrez et al., 2012*), psychological stress (*Daros et al., 2014*) or pharmacological treatments (*Verbeek et al., 2014*) showed negative judgement biases. By contrast, the study of positive judgement biases has produced inconsistent findings. A positive judgement bias has been associated with short-term (i.e., a few days or weeks) changes to housing conditions (*Matheson, Asher & Bateson, 2008*) (although not always confirmed; *Wichman, Keeling & Forkman, 2012*), long-term good care (*Briefer & McElligott, 2013*), with pharmacological treatment using diazepam or morphine (*Verbeek et al., 2014*), and with specific manipulations (i.e., tickling) (*Rygula, Pluta & Popik, 2012*). Rats (*Rattus norvegicus*) treated with oxytocin did not display a shorter latency to approach ambiguous trials compared to rats treated with saline solution. However rats (regardless of treatment), were significantly slower on the aversive trials compared to the ambiguous trials, and thereby indicating an overall positive bias (*McGuire et al., 2015*). Contrary to the predictions, positive judgement biases were also found when animals were released from short-term stressors (*Verbeek, Ferguson & Lee, 2014*). These inconsistencies could be attributed to the poor assessment of the rewarding or non-rewarding (punishment)

properties of the stimuli adopted or due to a lack of knowledge of animals' cognitive abilities to quantify and discriminate the ambiguous stimuli (*Baciadonna & McElligott, 2015*).

Interactions between humans and animals offer an interesting and valid way for testing the effects of induced positive emotions on judgement bias in animals. The quality of the relationship between human handlers and farm livestock has a large effect on animal wellbeing, productivity, and success in handling animals easily (*Tallet, Veissier & Boivin, 2005*; *Waiblinger et al., 2006*). For example, regular positive contact between humans and animals reduces fear reactions in animals (*Waiblinger et al., 2006*). Similarly, positive contact between humans and animals (e.g., petting/grooming) can generate an affinity for the stockperson with increased motivation to search and approach the caretaker (*Lürzel et al., 2016*) and calming effects (*Tallet, Veissier & Boivin, 2005*; *Coulon et al., 2015*).

Farm livestock might be particularly sensitive and responsive to positive interactions with humans due to their long history of domestication (*Zeder, 2000*; *Nawroth, Brett & McElligott, 2016*). In cattle (*Bos taurus*), grooming was associated with a reduction in cortisol levels, changes in cardiac activity linked with specific body parts (*Waiblinger et al., 2006*; *Schmied et al., 2008*), and also with changes in ear postures (*Proctor & Carder, 2014*). In lambs, gentle tactile contact with humans after a period of chronic stress was associated with a positive judgement bias (*Destrez et al., 2014*). With the aim of further exploring the use of specific human-animal interactions to induce positive emotional states in farm livestock, we investigated whether short-term strategies to boost the effects of routine positive care in goats would induce a positive judgement bias (*Briefer & McElligott, 2013*; Experiment 1). We predicted that grooming would induce a positive state and in turn an optimistic-like bias during a judgement bias test. The second aim of the study was to measure the physiological changes (i.e., heart rate and heart rate variability) and the behaviour associated with grooming (i.e., proximity to the experimenter), to determine whether this procedure was effective in inducing emotional changes (valence and arousal; Experiment 2).

## METHODS

### Experiment 1: judgement bias
#### Subjects and management conditions
The study was carried out at a goat sanctuary (Buttercups Sanctuary for Goats, http://www.buttercups.org.uk; Kent, UK). Nineteen adult goats (10 females and nine castrated males) of various breeds and age (Table 1) were tested from April to May 2014. Subjects were allocated either to an "experimental group" ($n =$ nine goats, five females and four castrated males) or to a "control group" ($n =$ 10 goats, five females and five castrated males). Goats that were used in this study had been at the sanctuary for a minimum of one year (range: 1–14 years). Employees and sanctuary volunteers provided routine care for the animals and therefore the goats were fully habituated to human presence and handling (*Briefer & McElligott, 2013*; *Baciadonna, McElligott & Briefer, 2013*). During the day, all goats were released together into one or two large fields that also provide shelters. During the night, they were kept indoors in individual or shared pens (average size $= 3.5$ m$^2$) with

**Table 1** **Characteristics of goats tested in the judgement bias experiment:** ID, breed, age, sex, treatment and rewarded side.

| ID | Breed | Age | Sex | Treatment | Rewarded side |
|----|-------|-----|-----|-----------|---------------|
| 1 | Mixed breed | 7 | Male | Control | Right |
| 2 | British Toggenburg | 11 | Male | Grooming | Left |
| 3 | British Toggenburg | 10 | Male | Grooming | Left |
| 4 | Golden Guernsey | 9 | Male | Grooming | Right |
| 5 | Pygmy Goat | 6 | Male | Control | Left |
| 6 | British Toggenburg | 3 | Male | Grooming | Right |
| 7 | Mixed breed | 14 | Male | Control | Left |
| 8 | Mixed breed | 9 | Male | Control | Right |
| 9 | Mixed breed | 9 | Male | Control | Left |
| 10 | British Alpine | 8 | Female | Control | Right |
| 11 | British Saanen | 10 | Female | Grooming | Right |
| 12 | British Toggenburg | 10 | Female | Grooming | Left |
| 13 | British Alpine | 10 | Female | Grooming | Right |
| 14 | British Saanen | 4 | Female | Control | Left |
| 15 | British Saanen | 4 | Female | Control | Right |
| 16 | British Toggenburg | 2 | Female | Grooming | Left |
| 17 | British Toggenburg | 3 | Female | Grooming | Left |
| 18 | Anglo Nubian | 8 | Female | Control | Right |
| 19 | Boer | 1 | Female | Control | Left |

straw bedding, within a large stable complex. Goats had *ad libitum* access to hay, grass (during the day) and water and were also fed with a commercial concentrate in quantities that varied according to their health and age.

### Treatment

Goats of the experimental group were gently groomed by one of the authors (LB) with a commercial animal brush. LB has been involved in research at the study site since 2011 and was therefore very familiar to the animals. Goats were familiar with the brush because this was occasionally and intermittently used by staff and volunteers at the sanctuary to remove dirt from their hair and not for inducing positive emotional states *per se*. Animals were groomed on the frontal and lateral part of the head and behind the horns and on the back (close to the base of the tail). These body parts were selected because animals at the sanctuary often scratch these same areas against trees branches or large boulders (L Baciadonna, pers. obs., 2011). The experimental group received five min of grooming before the training session, for nine days over two weeks. They also received five min of grooming with the experimenter before the test session, for two days. Therefore, in total, each animal received 55 min of grooming over 11 days. We expected grooming to induce a positive emotional state (*Schmied et al., 2008*; *Proctor & Carder, 2014*; *Destrez et al., 2014*; *Schmied, Boivin & Waiblinger, 2008*). The control animals were kept unconstrained adjacent to the experimenter for the same period of time as the goats in the experimental group (five min for nine days of training, plus two days of testing), but were not groomed.
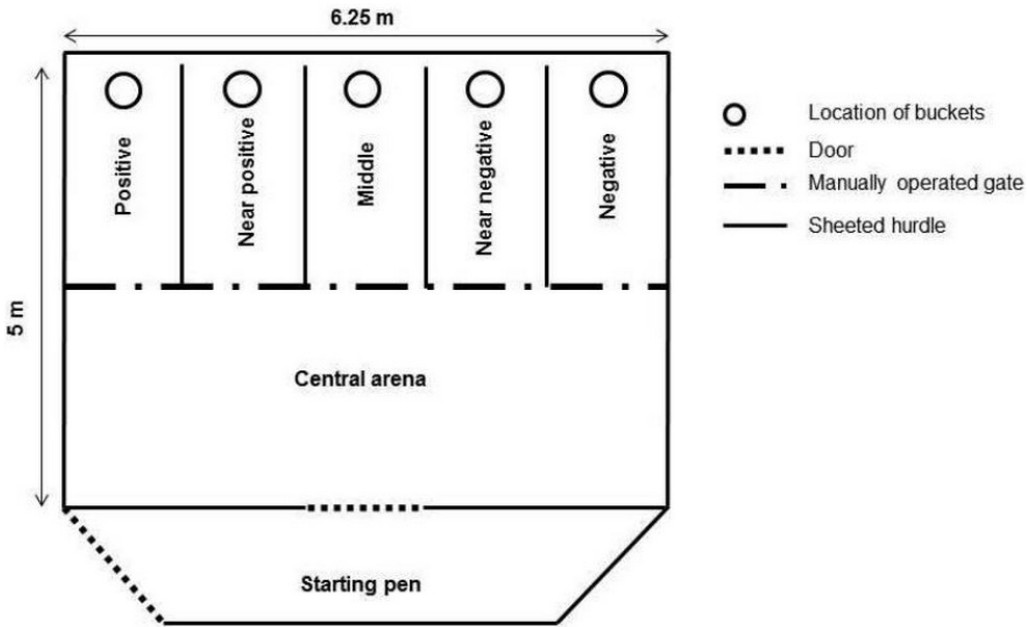

**Figure 1 Experimental apparatus.** Position of the positive corridor (right or left depending on the goats), the negative corridor (opposite direction), the three ambiguous corridors, the central arena and the start pen. The latency to reach the locations was measured (distance from the start pen to the beginning of each corridor).

## Experimental apparatus

An experimental apparatus (5 m × 6.25 m) was set up and placed in one of the fields that is part of the goats' normal daytime range (*Verbeek, Ferguson & Lee, 2014*; Fig. 1). It consisted of a start pen (5 m × 1.25 m) connected by a door to a central arena and five corridors (corridor length = 2.50 m, corridor width = 1.25 m) made of sheeted livestock fencing (height = 1 m). The central arena allowed opening or closing of a manually operated gate to provide access to the corridors. The choice of a specific corridor (either on the right or left side of the arena) was rewarded with a mix (approximately 70–80 g) of apples and carrots ("positive corridor"). The corridor at the opposite side of the arena was never rewarded ("negative corridor"). Three ambiguous corridors were positioned between the positive and negative corridors. One ambiguous corridor was positioned next to the positive corridor ("near positive"), one was positioned in the middle ("middle corridor"), and one next to the negative corridor ("near negative"). The ambiguous corridors were never rewarded in order to avoid associations between these locations and the presence of a food reward. A grey bucket with food (positive corridor) or an empty grey bucket (negative or ambiguous corridors) were placed at the end of the corridors. The buckets were covered with a plastic lid (8 mm thick) in order to reduce olfactory cues that could indicate the presence of food.

## Habituation

To familiarise the animals with the experimental apparatus, each goat was individually placed in the apparatus twice, for 12 min, over two consecutive days. Each session consisted

of two min in the start pen, followed by 10 min of exploration inside the arena. During the habituation phase, all five corridors were opened and a small quantity of food (mix of apple and carrots) was scattered in the enclosure to encourage exploratory behaviour. A grey bucket with a lid was placed in the middle of the central arena. This bucket was used also during the training and test phases. This allowed the goats to associate the grey bucket with the food reward and to practice how to remove the lid and retrieve the food.

### Judgement bias training

Nine goats (five females and four males) were trained to expect food on the right side (positive corridor; four goats from the experimental group and five from the control group) and 10 goats (five females and five males) were trained to expect food on the left side (positive corridor; five goats from the experimental group and five goats from the control group, Table 1). Goats were tested in random order. The experimental group (nine goats) was groomed for five min before starting the training procedure. The control group (10 goats) was also placed in the starting pen for five min before the training and kept adjacent to the experimenter. The ambiguous corridors (near negative, middle, and near positive) remained closed during this phase. Only one corridor at a time (either positive or negative) was open. During the first session of training all goats received two consecutive positive trials followed by two consecutive negative trials plus two additional trials where they were trained to reach the positive and negative corridors alternatively (six trials in total). This was in order to facilitate discrimination between the two locations. For the other training sessions, a pseudo-random order with no more than two consecutive positive or negative trials and with the same number of positive and negative trials per session was used (*Briefer & McElligott, 2013*) A significant shorter latency to reach the positive than the negative corridor was obtained for all goats on the second day of training (linear mixed-effects models: $p \leq 0.001$). At the end of each training day the average latency time to reach the positive and negative corridors was calculated. The training ended after nine days, when the latency to approach the positive corridor was on average less than 5 s and the latency to reach the negative corridor was more than 100 s.

### Judgement bias test

The test phase was conducted over two consecutive days. During each testing day, goats were tested over seven trials (i.e., one session). In particular, they were tested three times with the ambiguous corridor, two times with the positive corridor, and two times with the negative corridor. The positive and negative trials were repeated twice, as a reminder. The ambiguous corridors were opened in random order and were alternated with the positive and the negative corridors. Indeed, the ambiguous corridors were tested after the positive or after the negative corridor over the two days (*Briefer & McElligott, 2013*).

### Training and testing procedure

During the training and testing trials, the goats were individually brought to the start pen and groomed (only the experimental group) for five min. After grooming, the experimenter opened the gate of the appropriate corridor. The bucket was filled with food for a positive trial or we pretended to fill the bucket (making noise with food) for a negative or an

ambiguous trial. The bucket was subsequently covered with the plastic lid. Next, the start pen door was opened to allow the goat to enter the central arena. The experimenter waited for the goat to reach and cross the line and allowing the time to eat the food (positive corridor), or to reach and cross the line at the beginning of the corridor before returning to the start pen. A short inter-trial interval (<1 min) followed to prepare for the next trial. During each training and test session, the time from when the animal's two front legs passed the line on the gate at the entrance of the central arena to the time when they reached and crossed the line at the entrance of the target corridors with the two front legs was recorded. If the goat did not enter the central corridor from the start pen in 90 s, the door was closed and training/testing session continued. If the goat did not cross the line at the entrance of the open corridor, it was brought back to the start pen after 180 s and the training/testing session continued to the next trial. All sessions were recorded using a digital video camera placed behind the subject (Sony HDR-CX190E). The experimenter (who was not blind to the treatment) recorded the latency time directly. A second observer, blind to the experimental hypotheses, scored 20% of the total sessions to test the reliability of the latency times recorded (*Tuyttens et al., 2014*). The inter-observer agreement for latency time was high (Spearman rank correlation; $r_s = 0.976$; $p < 0.001$).

## Experiment 2: physiological effects of the grooming
### Subjects and management conditions
The general management conditions of the animals are described in 'Subjects and management conditions'. Ten goats (five females, five castrated male) were tested to assess the effect of grooming on the physiological level during December 2015. Only six goats used in Experiment 1 were available for Experiment 2 and therefore four goats were naïve when they participated in Experiment 2. Goats were tested twice on two non-consecutive days; once without being groomed with an experimenter close to the subject (control) and the second time being actively groomed for five min by the experimenter. The aims of Experiment 2 were to assess the physiological changes of both branches of the autonomic nervous system (sympathetic and parasympathetic) using heart rate, and to examine the activation of the parasympathetic system using heart rate variability (*Briefer, Tettamanti & McElligott, 2015*; *Von Borell et al., 2007*). The behaviour associated with grooming (i.e., proximity to the experimenter) was also recorded. These two different types of data (i.e., physiological and behavioural) allowed us to determine whether the grooming was effective in inducing emotional changes in valence and arousal.

### Treatment and physiological recordings
The goats were groomed with a commercial animal brush in one of the indoor pens where they were normally kept overnight. Goats were groomed on the frontal and lateral part of the head, the part behind the horns and on the back (close to the tail). The physiological parameters were recorded using a non-invasive device, fixed to a belt placed around the goat's chest (EC38 Type 3, BioHarness Physiology Monitoring System; Zephyr Technology Corporation, Annapolis, MD, USA).

Heart rate was measured using the BioHarness system. The week before the test commenced, a small patch of hair (7 cm × 15 cm) was clipped so that the heart rate

monitor worked more effectively. The BioHarness was also attached to the animal for a short period of time (five min) in order for habituation to occur. The habituation was conducted for a short period of time because the goats that participated in this study had previously experienced wearing the device during other research (*Briefer, Tettamanti & McElligott, 2015*; *Briefer, Oxley & McElligott, 2015*). The continuous ECG trace was transmitted online to a laptop (ASUS S200E) and stored using software (AcqKnowledge 4.4, BIOPAC System Inc) for later analyses. When the heartbeats were clearly visible on the ECG trace, 10 s sections ("start", "middle" and "end"; mean ± SD for each of the three sections: "start": 10.12 ± 0.68 s; "middle": 10.00 ± 0.71 s; "end": 10.06 ± 0.74 s) were selected and analysed. Heart rate and heart rate variability (i.e., root mean square of successive interbeat interval differences, RMSSD) were analysed from the ECG trace. To improve the quality of ECG trace, any electrical noise was removed by selecting Line frequency of 50 Hz (from AqcKnowledge > Transform > Digital Filters > Comb Band Stop). Baseline drift and movements artefact were also removed using a high pass filter at fixed cut off frequency of 1 Hz (from AqcKnowledge > Transform > Digital Filters > IIR > High Pass). The AcqKnowledge software provided the heart rate (beats/min) automatically. Individual intervals between heartbeats were also extracted to calculate RMSSD. All sessions were recorded using a digital video camera placed behind the subject (Sony HDR-CX190E). The total time that the experimenter actively groomed the goats was recorded during the grooming session.

## Data analysis

For Experiment 1, the average latency to reach the positive and negative location on each training day was calculated for each subject. For the testing phase, the latency to reach the locations over the two sessions was averaged for each goat (*Briefer & McElligott, 2013*). The latency data from the training and testing phases were analysed with linear mixed-effects models (Linear Mixed Effect Model (LMM); lmer function, lme4 library; *Pinheiro, 2000*) in R 3.2.2 (*R Development Core Team, 2013*). The linear mixed-effects model analysis allowed us to examine the following variables: "treatment" (groomed vs. control), "location" (positive, negative, near positive, middle and near negative), "age," "training day," and "side" (reward side) as fixed effects. The identity of the goats was included as a random factor to control for repeated measurements of the same subjects. The LMM allows the elimination of the non-significant variables considered in the model if it does not cause any significant reduction in goodness of fit of the model using a standard model simplification procedure. The two models with and without each term, both fitted with the maximum likelihood method (ML), were compared using a likelihood ratio test. The results are presented after model simplification and with restricted maximum likelihood method (REML). When an interaction effect was found, further posthoc comparisons were performed using LMM including control factors that remained in the final models. Bonferroni correction was applied to the post-hoc comparisons. Q–Q plots and scatterplots of the residuals of the model were inspected visually. In order to meet the assumptions the latency times were transformed using a reciprocal transformation ($1/X_i$).

The data for Experiment 2 were analysed using a LMM that allowed us to investigate for effects of the following variables: "treatment" (groomed/control), "section" (the part selected for the HR and HRV; Start, Middle and End) and "sex" as fixed effects. The same standard elimination procedure used for the judgement bias experiment was applied as previously described. The identity of the goats was included as a random factor to control for repeated measurements of the same subjects. Q–Q plots and scatterplots of the residuals of the model were inspected visually to verify the test assumptions.

### Ethical note

Animal care and all experimental procedures were conducted in accordance with the Association for the Study of Animal Behaviour (ASAB) guidelines (*Association for the study of animal Behaviour, 2016*). The study was approved by the Animal Welfare and Ethical Review Board of Queen Mary University of London (25042014FdQMUL). The tests were non-invasive and lasted less than 10 min (including the preparation time for adjusting the belt around the chest of the subject and the grooming treatment) for each animal. Behaviours indicating stress (frequent vocalizations and rapid movements away from the experimenter) were monitored throughout the exposure to grooming. None of the goats displayed behavioural signs of stress during the experiment.

## RESULTS

### Experiment 1: judgement bias training

An interaction effect between "training day" and "location" was found (LMM: $X^2_{(1)} = 202.35$, $p < 0.001$). Post-hoc analyses, after Bonferroni correction ($p \leq 0.01$) indicated that the goats learned the task on the second day of training ($p < 0.001$). Goats reached the positive corridor faster (latency mean $= 15.60 \pm 4.96$ s) than the negative corridor (latency mean $= 27.61 \pm 7.04$ s; $n = 19$ goats, Fig. 2). An interaction effect between "sex" and "location" was also found (LMM: $X^2_{(1)} = 6.97$, $p = 0.008$). Post-hoc analyses after Bonferroni correction ($p \leq 0.01$), revealed that females ($p < 0.001$) and males ($p < 0.001$) approached only the non-rewarded corridor differently (LMM: $X^2_{(1)} = 4.51$, $p = 0.03$; rewarded corridor LMM: $X^2_{(1)} = 0.64$, $p = 0.42$). The difference on the non-rewarded corridor was not retained after correction ($p > 0.01$). The other terms included in the initial model, namely "treatment" (LMM: $X^2_{(1)} = 0.21$, $p = 0.64$), "side" (LMM: $X^2_{(1)} = 1.22$, $p = 0.26$) and "age" (LMM: $X^2_{(1)} = 2.44$, $p = 0.11$), and the interaction terms ($p \geq 0.05$) did not significantly affect the latencies time during the training phase.

### Experiment 1: judgement bias test

The model selection procedure for the testing sessions revealed an effect of location on the general latencies (LMM: $X^2_{(1)} = 89.55$, $p < 0.001$), with goats reaching the positive corridor faster than the negative one, and the ambiguous corridors with intermediate latencies (Fig. 3). There was no interaction effect between the "treatment" and "location" on the latency to reach the five corridors (LMM: $X^2_{(4)} = 4.10$, $p = 0.39$). A weak interaction effect between "treatment" and "sex" was found (LMM: $X^2_{(1)} = 3.63$, $p = 0.056$). Post-hoc analyses, after Bonferroni correction ($p \leq 0.01$), revealed that females were slower than

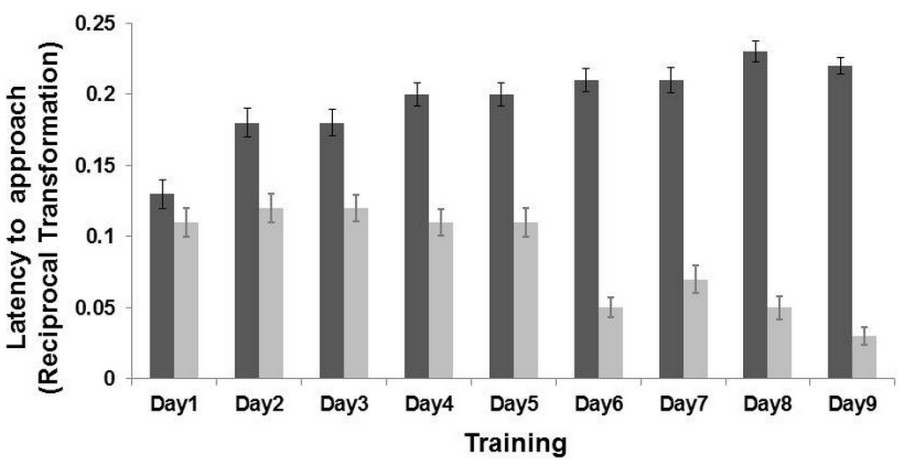

**Figure 2** **Results of the training phase.** Latency (Mean ± SE) to reach the positive location (dark grey bar) and the negative location (light grey bar) during the nine days of training. The latency time was transformed ($1/Xi$), and therefore higher latency times indicate faster approaches and vice versa. There was an interaction effect between training day and locations (LMM: $p < 0.001$).

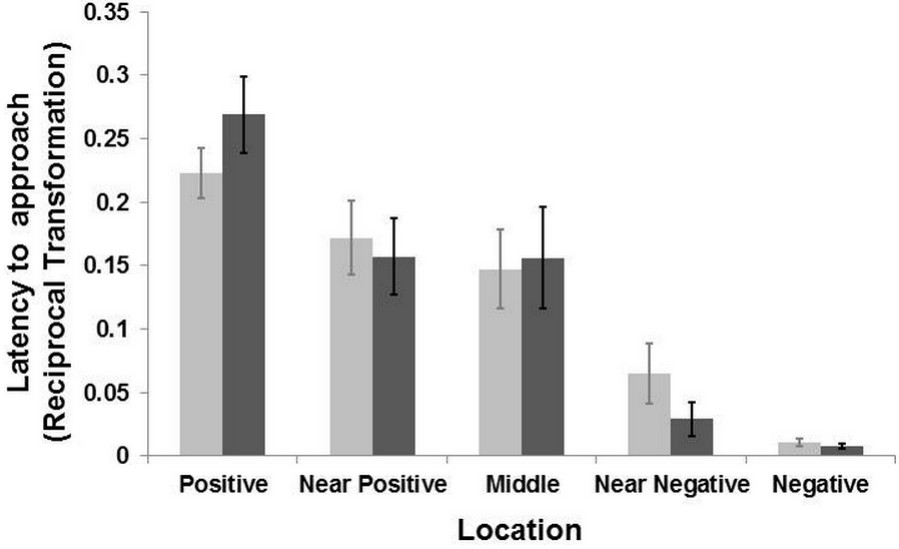

**Figure 3** **Results of the judgement bias experiment.** Latency (Mean ± SE) to reach the five locations during the two days of test for the groomed group (dark grey bar) and the control group (light grey bar). The latency time was transformed ($1/Xi$) and therefore higher latency times indicate faster approaches and vice versa. There was a general effect of location (LMM: $p < 0.001$) but no interaction between locations and treatment (LMM $p > 0.39$).

males overall (LMM: $X^2_{(1)} = 6.29$, $p = 0.01$), regardless of treatment condition. Males reached the five corridors faster (mean latency = 51.74 ± 10.19 s, $N$ goats = 9) than females (mean latency = 62.37 ± 10.17 s; $N$ goats = 10). After the Bonferroni correction, the effect of "treatment" was not retained ($p > 0.05$). An effect of "age" on the latency to reach the locations was also found (LMM: $X^2_{(1)} = 5.53$, $p = 0.01$). Goats aged one to seven year old reached all corridors faster than those aged 8–14 years old (mean latency = 42.32

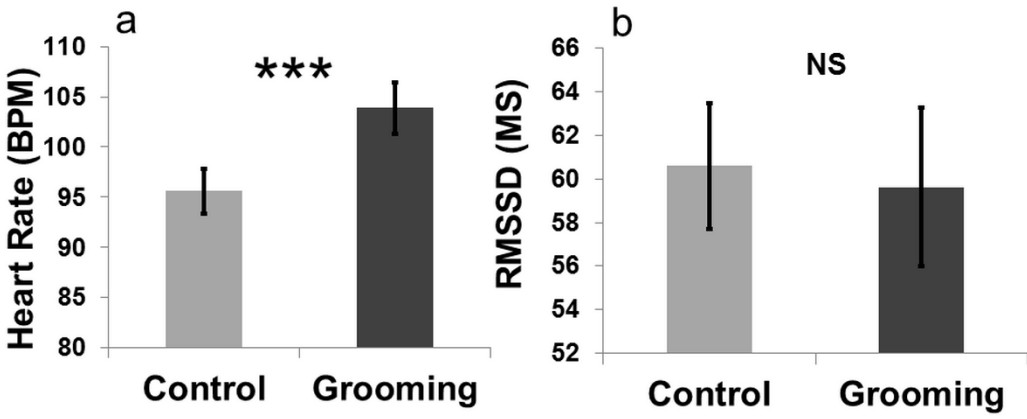

**Figure 4** (A) Heart rate (measured on the same animals) increased when goats were groomed compared to when the same animals were kept close to the experimenter without being groomed. (B) Heart rate variability (RMSSD) was not significantly different in the grooming compared to the control treatment.

$\pm$ 9.38 s, $N$ goats $= 8$; mean latency $= 68.25 \pm 10.20$, $N$ goats $= 11$). To summarize, there was no effect of grooming on the approach latencies to the five corridors. However, females were slower than males when approaching the corridors. An effect of age was found, with younger subjects faster than older ones

## Experiment 2: physiological activation during the grooming

An effect of "treatment" on heart rate was found (LMM: $X^2_{(1)} = 11.63$, $p < 0.001$). Heart rate was higher when the goats were groomed (mean BPM $= 103.90 \pm 2.58$) compared to the control (close to the experimenter without being groomed; mean BPM $= 95.59 \pm 2.27$, Fig. 4). The other terms included in the initial model, namely "sex" (LMM: $X^2_{(1)} = 0.06$, $p = 0.80$) and "section" (LMM: $X^2_{(2)} = 1.47$, $p = 0.47$), and the interaction terms ($p \geq 0.05$) did not significantly affect heart rate. "Treatment" had no effect on heart rate variability (RMSSD; LMM: $X^2_{(1)} = 0.04$, $p = 0.83$). The other terms included in the initial model namely "sex" (LMM: $X^2_{(1)} = 0.78$, $p = 0.37$) and "section" (LMM: $X^2_{(2)} = 4.59$, $p = 0.10$), and the interaction effect ($p \geq 0.05$) did not affect the heart rate variability. The videos showed that goats did not avoid being groomed (i.e., they did not move away when the experimenter approached) for most of the time (mean $= 287 \pm 10.50$ s; 95.66% of the total amount of time allowed). Heart rates increased when the goats were groomed compared to when they were kept inside the pen with the experimenter without engaging in any contact (Fig. 4).

## DISCUSSION

Human-animal interactions can have huge impact on the emotional lives of animals (*Waiblinger et al., 2006*). Investigating this topic can provide valuable information to promote positive emotions and psychological wellbeing in animals (*Boissy et al., 2007*). In this study, we used short-term positive human-animal interaction (i.e., grooming) to attempt to induce positive emotional states in goats (*Coulon et al., 2015*; *Schmied, Boivin & Waiblinger, 2008*; *Lürzel et al., 2015*). We hypothesised that grooming would induce

positive emotional states, which in turn would lead to an optimistic-like bias. We found no significant differences in the judgement of ambiguous stimuli between goats that had been groomed and goats that had not received this treatment. However, a significant effect of age on the latencies to reach the corridors was revealed, with younger goats faster than older goats in choosing a corridor. These findings may indicate that grooming did not induce strong enough positive emotional states in goats, or that the performance in the judgement bias test was not influenced by positive emotions. In the second experiment we found that the heart rates of the goats increased as a result of being groomed, and also, that the animals accepted the grooming most of the time. This suggests that the treatment was perceived by the animals, but could not be detected during the judgement bias test. The use of the judgement bias test in farm animals is controversial and has produced discordant findings (*Roelofs et al., 2016*; *Baciadonna & McElligott, 2015*). More research is needed to identify effective strategies to induce positive emotions and to develop assessment tools able to detect emotional changes, especially positive ones (*Désiré, Boissy & Veissier, 2002*; *Boissy et al., 2007*).

To test the effectiveness of the manipulation we performed an experiment in which physiological activation was recorded in two treatments (i.e., control, with no contact with the experimenter, and grooming). We found that heart rates were higher during grooming compared to the control treatment. In combination with the behavioural finding that animals accepted being groomed for most of the time (95.66%) by the experimenter, this suggests that the grooming not only induced an emotional change in arousal, but also a change that was perceived as positive. This supports the hypothesis that grooming was effective in inducing an emotional change, and that the judgement bias task might have not been able to detect this change.

Heart rate and heart rate variability measurements are good indicators of emotional arousal and valence when used in combination with other parameters, such as behavioural responses and postures (*Briefer, Tettamanti & McElligott, 2015*; *Reefmann, Wechsler & Gygax, 2009*; *Zebunke, Puppe & Langbein, 2013*). For example, sheep (*Reefmann et al., 2009*) exhibited a higher inter-heartbeat interval (R–R interval) and higher heart rate variability when groomed compared to when they were standing in their home pen or in isolation. Lambs regularly stroked in early age and with strong bonds with humans also showed lower HR in the presence of their caregiver and while groomed, and higher RMSSD when compared with lambs that were not stroked (*Coulon et al., 2015*). In cattle, stroking and gentle human voices were associated with reduced heart rate following an aversive event (veterinary procedure; *Waiblinger et al., 2004*). However, in dogs, the RMSSD did not increase as expected whilst experiencing a positive situation (palatable food) (*Zupan et al., 2015*; *Travain et al., 2016*). The activation of the vagal tone, in dogs, has been suggested to occur when animals experience a further increase in the positive emotion that they were already experiencing (*Zupan et al., 2015*). In our case, it is most likely that goats had not experienced the grooming for a long enough period of time. This might have prevented them from developing a specific bond with the experimenter and from showing changes in heart rate variability as a consequence.

Similarly, short-term exposure to positive interactions (i.e., five min over 11 days; 55 min in total over six weeks) may not have been strong enough to further improve and boost the positive emotional states and experience of the goats that we used (*Briefer & McElligott, 2013*; *Schino et al., 2016*). Goats at our study site are kept in generally excellent conditions (i.e., according to the DEFRA Codes of Recommendation for the Welfare of Goat; *Briefer & McElligott, 2013*; *Department for Environment, 2006*) and they are used to experience positive interactions with people. These conditions are not comparable to those of laboratory animals or to the situations of chronic stress to which farm animals are normally exposed before experiencing a positive event (e.g., gentle tactile contact with a human) in a judgement bias test study (*Destrez et al., 2012*). In addition, although we selected the body parts that were groomed because animals at the sanctuary often scratch these against tree branches or large boulders (L Baciadonna, pers. obs., 2011), these parts might have not been appropriate to respond to a gentle tactile stimulation (*Schino, 1998*). Previous research has indicated the importance of selecting specific body parts for the grooming to be effective, such as regions touched during social behaviour (*Schmied et al., 2008*; *Proctor & Carder, 2014*; *Schmied, Boivin & Waiblinger, 2008*). For example, cattle groomed on the ventral part of the neck showed less avoidance behaviour compared with cattle groomed in the lateral side of the chest or withers (*Schmied, Boivin & Waiblinger, 2008*). The efficacy of the grooming could be linked to the person who performed the manipulation (*Schmied et al., 2008*). In order to generalise the results, it would useful to use more than a single experimenter to perform the grooming.

We found that age affected the overall performance in the judgement bias test. Younger animals approached the corridors faster than older ones (i.e., 1–7 year old goats faster than those aged 8–14 years old). This effect of age was not found during the training phase and suggests that age differences are unlikely to be related to physical effects. A faster approach during the judgement bias test could be due to impulsivity, defined as incapability to refrain from a motor response (*Weafer & De Wit, 2014*). Impulsivity has been associated with young ages in humans and non-human animals (*Andrzejewski et al., 2011*; *Burton & Fletcher, 2012*) and declines gradually with increasing age (*Laviola et al., 2004*; *Doremus, Varlinskaya & Spear, 2004*). Thus, age could affect the use of specific coping strategies in unpredictable/new situations. To avoid any potential confounding effect associated with impulsivity and motivation, the use of Active Choice has been suggested as an alternative to the Go/No-go task (*Roelofs et al., 2016*). In an Active choice task, the animals must perform an action directed towards both the positive and negative stimuli, instead of simply displaying an absence of response to the negative stimuli.

## CONCLUSION

In conclusion, we did not find evidence of a positive judgement bias after goats had been groomed. To exclude that these results were due to the inefficacy of the grooming to induce an emotional change, we performed a second experiment in which physiological parameters were recorded. We found an increase in heart rate when goats were groomed suggesting that they were sensitive to the treatment. Thus, the grooming potentially induced an emotional

change but this was not detected during the judgement bias test. The performance in the judgment bias test was influenced by the age of the animals. Our findings demonstrate the importance of combining behavioural, physiological and cognitive factors to assess the emotional states experienced by animals. In addition, taking into account individual characteristics of the animals (e.g., age, sex and personality, *Briefer, Oxley & McElligott, 2015*) and clarifying which emotional states are identifiable by a judgment bias paradigm could increase the effectiveness of cognitive bias paradigms to assess emotional valence (*Baciadonna & McElligott, 2015*).

## ACKNOWLEDGEMENTS

We thank Eric Romero Gonzalez for helping with data collection, Marie-Sophie Single for video analysis, and Caroline Spence and the editor and reviewers for their helpful comments. We thank Bob Hitch, Gower McCarthy, Samantha Taylor and all the volunteers at Buttercups Sanctuary for Goats (http://www.buttercups.org.uk) for their excellent help and free access to the animals.

### Funding
The authors received no funding for this work.

### Competing Interests
The authors declare there are no competing interests.

### Author Contributions
- Luigi Baciadonna, Christian Nawroth and Alan G. McElligott conceived and designed the experiments, performed the experiments, analyzed the data, contributed reagents/materials/analysis tools, wrote the paper, prepared figures and/or tables, reviewed drafts of the paper.

### Animal Ethics
The following information was supplied relating to ethical approvals (i.e., approving body and any reference numbers):

Animal Welfare and Ethical Review Board of Queen Mary University of London (25042014FdQMUL).

### Data Availability
The raw data has been supplied as a Data S1.

### Supplemental Information
Supplemental information for this article can be found online at http://dx.doi.org/10.7717/peerj.2485#supplemental-information.

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
