# Peer review of "Judgement bias in goats (Capra hircus): investigating the effects of human grooming"

_PeerJ, doi:10.7717/peerj.2485_

## Round 0.1 · original submission · Major Revisions

I was very fortunate to receive thorough reviews from three expert reviewers. Although two of the reviewers indicated only minor revisions were needed, they each identified several serious issues that needed revision or clarification. The other reviewer had far more serious concerns; the largest concern being about the lack of difference between the experimental and control groups. If you are testing the judgement bias at the moment of decision and only the experimental group had just received grooming, I think you may be justified to distinguish the two groups even if the control group did sometimes receive grooming throughout the day, but Reviewer 1 and 3's point about the familiarity of the experimenter doing the grooming is also important. It is possible that the experience was more aversive than rewarding for the goats. I am also less concerned by the fact that you did not reward in the ambiguous corridors because the goats did not have that experience until testing, which is standard practice for such paradigms. I am concerned with the brief grooming period and the inability to determine with certainty if it was a pleasurable experience for the goats. I think your paper could really benefit from some additional conditions to rule out alternative reasons for the lack of an effect. That is, you could lengthen the grooming session, have the grooming performed by a familiar caretaker, and remove grooming or interactions entirely for the control group. Adding such data would increase the likelihood of the paper being accepted for publication. At the very least, you will need to convincingly rebutt the concerns of the reviewers.
How did you test to determine that response latencies to the reward and non-reward corridors were statistically distinct for individuals during training?

Could you please say something about the power of your model to detect significant effects?

Your effects and lack thereof are similar to those obtained by McGuire et al. (2015, Frontiers in Comparative Psychology) using a modified CPP apparatus and testing the effects of OT on bias in rats. You might consider citing that paper here. Your MS contributes to a growing literature finding no or ambiguous results in a judgement bias paradigm.

Your data is not continuous between conditions so you should be presenting it using bar graphs rather than line graphs.

Lastly, you must obtain formal approval for the studies before the studies can be published in PeerJ. A letter of request to obtain approval is not sufficient.
Thanks for submitting such an interesting contribution to PeerJ.

·

Basic reporting

The article was well written. No Comments.

Experimental design

How familiar was the experimenter to the goats? Could it have been that using a familiar caretaker would have produced more of a behavioral response due to the fact that the goats were somewhat uneasy around the experimenters (despite being habituated to the act of grooming)? I would have been interested in how they would have performed had their regular caretakers been the ones performing the grooming sessions.

In line 177, please describe how you determined significance.

In the second experiment, why wasn't the order of the sessions counterbalanced? Why always start with the non-grooming session and end with the grooming session?

Were the goats habituated to the heart rate monitoring equipment before the experimental sessions?

Why did you settle on 5 minutes of grooming? As you stated in line 355, it is possible that this was not an adequate amount of time to achieve the changes in affect that you were looking for. How long would a true grooming session last? If you had groomed more areas of the body you may have increased your chances of influencing their emotional state as you would have potentially groomed more sensitive areas (which you talked about in your discussion) and it would have increased the amount of time in which they were experiencing the grooming,

Validity of the findings

You stated in lines 375-380 that heart rate can be an ambiguous indicator- you give examples of cases in which an increase in HR is interpreted as a positive indicator and also cases in which a decrease in HR is interpreted as a positive indicator. How can you be sure that your current interpretation is correct? Couldn't you have theoretically interpreted any change in HR as "positive" then? I know that HR can also be an indicator of negative affect (fear) so this seems troubling.

·

Basic reporting

1. Overall, the manuscript is written in a clear, professional language.
2. The introduction follows a clear structure, explaining important concepts and leading to a well-defined research question. The relevance of this research question is indicated by a framework of well referenced literature.
3. Lines 48-50 requires rephrasing, sentence now states that because human perceptions/judgments are influenced by emotions, human studies can inform the development of animal paradigms. The sentence was likely meant to state that animal studies of judgment bias are based on paradigms developed for human studies.
4. In line 51, the mention of eating disorders seems irrelevant as it is not explained how those would be linked to emotional states (whereas depression and anxiety are clearly related to negative emotional states).
5. Lines 52-54 are very general and require rephrasing/specification. This is attempted by an example in lines 54-57, describing responses to expectance violation in sheep. However, the relevance of this example to judgment bias studies is unclear and seems unsuitable to take readers from human studies of judgment bias to judgment bias tasks in animals.
6. In lines 58-59, attention bias is provided as an example of a process studied using judgment bias paradigms. This is untrue, attention bias is a separate form of cognitive bias. It could be stated that cognitive bias paradigms can be used to measure judgment, memory and attention bias. See also Roelofs et al. 2016 for a recent review of judgment bias tasks and their application to animals.
7. In lines 69-74 examples of judgment bias studies aimed at positive emotional states are well described. Perhaps also of interest is a study with rats performed by Rygula et al. (2012), where a clear behavioural indication of positive emotional state (laughing when tickled) correlated with a more positive judgment bias.
8. The structure of the manuscript conforms to PeerJ standard and raw data are supplied.
9. Figures are relevant, high quality, with prober labels and descriptions. For figure 3, it could be added that besides a lack of interaction effect, also no general condition effect was found.
10. Table 2 is very unclear and does not receive a lot of attention in the manuscript. Omissions are presented as total number of omissions for all goats combined. However, it would be of interest how many goats are responsible for the omissions (e.g. there 94 omissions recorded for positive training trials, does this number consists of all goats making several omissions or a few goats hardly participating at all?).
11. Raw data are supplied.

Experimental design

12. The experimental design describes original primary research within the scope of the journal.
13. The research question is well defined and relevant. It explores an area of judgment bias research which requires attention, namely the measuring of positive affective states.
14. Section 2.1.3 described the treatment of the experimental group. Goats were brushed for 5 minutes during training and testing days. However, it is also stated that goats were already familiar with the brush because it is routinely used by staff at the sanctuary where they are housed. This leads to the question: is there truly a difference between experimental and control group if both groups are routinely brushed outside of training/testing conditions?
15. Section 2.1.4 describes the experimental apparatus and testing design. During testing, “ambiguous corridors were never rewarded in order to avoid associations between these locations and the presence of a food reward.” This means that ambiguous corridors have the exact same outcomes as the negative corridor. It seems plausible that the goats therefore associate the ambiguous corridors with the negative outcome (see also ‘loss of ambiguity within a small number of testing trials’ in Roelofs et al. 2016).
16. In section 2.2.2, it is stated that “not all goats (6/10) used in experiment 1 were available for experiment 2.” This is unclear, are 6 of the animals in experiment 2 ‘new animals’?
17. In section 2.3, lines 254-255, what is the difference between the fixed effects ‘sequence’ and ‘side’?
18. In section 2.3, line 270, it states that ‘sex’ was used as a fixed effect, this is not mentioned for experiment 1.
19. Throughout the methods and results, the terms ‘treatment’ and ‘condition’ appear to be used interchangeably. Please use only one of the terms, to improve clarity.
20. Investigation was performed to a high ethical standard.
12. The experimental design describes original primary research within the scope of the journal.
13. The research question is well defined and relevant. It explores an area of judgment bias research which requires attention, namely the measuring of positive affective states.
14. Section 2.1.3 described the treatment of the experimental group. Goats were brushed for 5 minutes during training and testing days. However, it is also stated that goats were already familiar with the brush because it is routinely used by staff at the sanctuary where they are housed. This leads to the question: is there truly a difference between experimental and control group if both groups are routinely brushed outside of training/testing conditions?
15. Section 2.1.4 describes the experimental apparatus and testing design. During testing, “ambiguous corridors were never rewarded in order to avoid associations between these locations and the presence of a food reward.” This means that ambiguous corridors have the exact same outcomes as the negative corridor. It seems plausible that the goats therefore associate the ambiguous corridors with the negative outcome (see also ‘loss of ambiguity within a small number of testing trials’ in Roelofs et al. 2016).
16. In section 2.2.2, it is stated that “not all goats (6/10) used in experiment 1 were available for experiment 2.” This is unclear, are 6 of the animals in experiment 2 ‘new animals’?
17. In section 2.3, lines 254-255, what is the difference between the fixed effects ‘sequence’ and ‘side’?
18. In section 2.3, line 270, it states that ‘sex’ was used as a fixed effect, this is not mentioned for experiment 1.
19. Throughout the methods and results, the terms ‘treatment’ and ‘condition’ appear to be used interchangeably. Please use only one of the terms, to improve clarity.
20. Investigation was performed to a high ethical standard.

Validity of the findings

21. Data is robust, statistically sound and controlled.
22. In lines 350-353 it is concluded that grooming did not induce a strong enough positive emotional state, or the judgment bias test was not influenced by positive emotional state. However, several other explanations appear possible given the experimental design. It seems that there could have been no difference in emotional state between experimental and control groups (as brushing was also used in regular care for the goats). Also, it is possible that the goats associated the ambiguous corridors with the negative treatment of no reward, resulting in a lack of optimistic responding.
23. Lines 359-361 are repetitions of lines 350-353.
24. Lines 365-370 state that grooming induced a change in emotional arousal, which was not detected by the judgment bias task. However, are judgment bias tasks meant to detect emotional arousal? As described in Mendl et al. 2009, judgment bias is a result of the valence of the emotional state of an animal, not the arousal. Therefore, why would it be expected that emotional arousal would have been detected using judgment bias testing?
25. In lines 395-403, the lack of measurable positive emotional state in the judgment bias task is explained as a result of possibly grooming the wrong body part of the goats. However, the heart rate responses indicate that grooming did indeed produce an emotional response. This makes it seem unlikely that reproducing the experiment with different groomed parts will lead to better results.
26. In lines 407-409, it is stated that an incapability to inhibit a motor response was responsible for the found age effect during judgment bias testing. Suppression of response is required for go/no-go judgment bias tasks. Perhaps using a go-go or active choice paradigm would be more suitable when testing animals with differences in impulsivity/motivation, as is likely when age of study subjects is so varied. This could be suggested for future research.
27. A sex effect was mentioned in the results section, but was not discussed.

Additional comments

Overall, it seems that the main flaw in the experimental design is a lack of difference between treatment and control group.

·

Basic reporting

The authors present data showing that a cognitive bias test of emotional valence does not detect the effect of brief periods of grooming from a human experimenter even though heart rate was found to increase. I enjoyed this manuscript and expect it to be a sound and valuable addition to several literatures including animal welfare, human-animal relations, and emotion. There appear to be some errors in defining and reporting the results that need to be corrected and I believe the manuscript would benefit from clarifying and expanding upon a few analyses and conceptual points.
* * *
Title & Abstract:

The title is a bit vague given that grooming could refer to self-grooming, allgrooming, or (as was the case) human grooming and, moreover, that it can last for a wide range of times. I would suggest clarifying that the grooming under study was from a human and was quite brief.

The order of events and methodology is unclear in abstract—perhaps begin the 3rd sentence with “During training, the experimental group…” and make a similar addition to the sentence on the control group as well.

The ad hoc and speculative nature of the age effects don’t warrant mention in the abstract. It may have nothing to do with impulsivity, but rather physical wear-and-tear. This alternative explanation should be addressed in the discussion.

Though the authors mention it in the body of the manuscript, they do not clarify in the abstract that cognitive bias is not designed to detect changes in emotional arousal, but rather emotional valence. Instead of discussing the possible impulsivity of the younger goats, I suggest contextualizing and clarifying the main results: it is unclear whether the lack of an effect in cognitive bias is due to (i) the failure of grooming to induce a positive mood or (ii) the failure of cognitive bias testing to detect brief changes in positive mood. What is clear, however, is that the grooming increased arousal and, as the authors argue elsewhere, since the goats voluntarily remained with the experimenter, it is unlikely that the valence of the arousal was negative.

As the authors mention in the discussion, most previous studies of handling effects were carried out in facilities where the animals would have received almost no positive interactions with humans. The different baseline in the present study (goats received supreme care and regular positive human-animal interactions) seems highly relevant to contextualize the results and overall contribution of the study.

Ending with a more general broad impact/relevance/contribution of the study sentence would be helpful.


Introduction:

Please define tachycardia


Methods:

Authors write that food varied in relation to state—in relation to what state? Health? Reproductive cycle?

Line 165: I think the authors mean The location of “the positive and negative corridors were counterbalanced…” ? Please clarify.

Line 171: “All goats received two positive trials followed by two negative trials during the first session of six training trials to facilitate discrimination between the two corridors.” I’m a little confused by this sentence. Perhaps the term “trial” is used to refer to two different things (a single instance of allowing the goat to choose to approach and the daily session of training)? Please clarify.

Line 185: I don’t see how testing ambiguous arms after a positive or negative one eliminates the effect of the previous trial, it just makes the effect of the previous trail different that if the previous trial had been an ambiguous one.

Please note and/or comment on experimenter not being blind to conditions.

The abbreviation LMM is not defined.

Sequence is defined in the text as food received on the left or right side of the area, but from the raw data, data analysis and graphs, I think sequence means whether a location was rewarded vs. not—essentially the same as the definition of the “location” (but without the mid-locations). Beyond fixing the definition in the methods, I would suggest removing the term sequence and just sticking with one term to refer to the rewarding properties of the location and an other to refer to the side on which the reward was located (left or right). It might also help to clarify that by side you mean left vs. right and not rewarded vs. not. These changes should be applied throughout manuscript—e.g. you refer to sequence in Fig. 2, which according to the current definition in the methods would suggest that the goats became more behaviorally lateralized (preferring left over right locations, for example) over time.


Figures:

All figures: What are the error bars representing?

Figure 3: Jitter points (add a small amount of noise to the x-axis value) to make error bars not overlap

Experimental design

The experiment was conducted with a single groomer. It is possible that different groomers would have had different effects (better or worse) on the goats’ emotional state, so it is somewhat problematical to generalize to the effect of human-goat grooming in general. The authors should comment on this limitation.

How well acquainted were the goats with the experimenter? Do the authors think level of familiarity would influence the effect of grooming?

How might the response demand and reward of food interact with the study manipulations? It is possible that the goats in the grooming condition were in a more positive mood, but that because of the brief grooming they were more relaxed and/or focused on social relationships rather than pursuing a possibility of food in the ambiguous cues (perhaps even because the grooming was so brief, they wanted more grooming rather than to move on to food unless it was a sure bet), whereas the goats in the control condition were getting bored/frustrated with waiting in the pen and were thus more eager to start exploring ambiguous cues in the apparatus even though they were in a worse mood. Not unique to this study, but still a limitation is that the positive and negative conditions were not equivalent in terms of response: the positive condition required a response whereas the negative condition required a non-response. Thus, rather than pure emotion, the test is confounded with relaxation/agitation, activity, energy surplus, food motivation, boredom (seeking out any stimulus regardless of expected valence). These motivational differences could have obscured the emotional state of the animals.

In the discussion (around line 361+) and throughout the manuscript, the authors often imply that the physiological measures will help clarify whether the goats were in a positive state or not. As they mention elsewhere, strictly speaking the heart rate increase is only a sign of arousal. Interpreting it as arousal of positive valence relies on the observation that the goats remained with the experimenter voluntarily. Since this behavioral measure is the crux of the authors’ argument, they need to spend more time describing this behavior and how it was scored in the methods and preparing the readers for this logic in the introduction (e.g. some of the paragraph starting on line 371 should appear in the introduction). They also should check throughout the manuscript to make sure they are very clear that the physiological measure on it’s own only tests arousal (and not valence) and therefore requires some contemporaneous behavioral measure for making inferences as to mood.

Validity of the findings

Can the authors comment on why time at sanctuary was not included in the models--seems like relative newcomers would respond differently to humans that those who had been there for longer.

Was day entered as a continuous or categorical variable? From the models and data, it appears to be continuous, but from the graph it appears to be categorical (each day has its own error bar rather than a linear trend with confidence intervals, as would be expected for a continuous variable).

Was position entered as a continuous or categorical variable? (Data and graphs indicate categorical, but the model only reports one degree of freedom (line 307), which indicates a continuous variable)

Thank you for reporting the participation and rate and behavior of the goats. However, I have several questions regarding the trials in which the goats did not enter the test apparatus—it seems that disengagement from the test (not leaving the start box and/or not entering the arms) is a valid if unanticipated response option and should be considered. It appears that by the test phase of the negative location, they were more often refusing to go into the corridor than at any other point in the study, which makes sense and indicates that a refusal to participate may be a meaningful response. Also, it is unclear to me what happened if a goat refused to participate. The authors write that the door was closed (or the goat was brought back to the start pen) and training continued. How did it continue? Did you re-run the same trial or did you move on to the next trial in the schedule? If you moved on, did you give the goat a latency of 90 (or 180) for the ‘refused’ trial? Even if you didn’t move on (i.e. re-ran the same trail), these “non-responses” still seem like valid data points. The authors might want to consider a survival analysis—the trials during with the goats refused to participate would remain in the data set as “censored” values along with the latencies for the trails in which they did participate. Other things to check regarding the refusal behavior is whether there were differences by condition (grooming vs. control) and whether there were individual differences and/or sex and age differences in refusing to participate. (These questions could be addressed with a logistic regression of refusal yes/no 1/0)

The authors report a sex by sequence (reward) interaction, but only discuss the main effect of sequence (reward)—the interaction indicates that the magnitude of the difference between reward (pos. vs. neg.) latencies was different for different sexes. From the values reported, it appears that females were more sensitive to reward difference than males—why might this be?

Similarly, the sex by condition interaction was only explained in terms of main effect of sex, but the interaction indicates that the effect of condition was different for each sex.

---

## Round 0.2 · Minor Revisions

Decision on Judgement bias in goats (Capra hircus): Investigating the effects of human grooming

Thank you for your detailed response to the reviewers’ comments. I believe you have made a good case for your experimental approach and the resulting MS is clearer. However, you appear to have exerted more effort responding in your response letter relative to changes made in the actual manuscript. Because readers will not have the benefit of the review process please be sure to expand the discussion within the text more carefully. I am inviting you to undergo another set of minor revisions to accomplish this. What follows below are only minor suggestions for clarity but they should not be taken as exhaustive. Please incorporate other details from your response letter.

Please write out in full numbers less than 10 where appropriate.
If younger goats approached all locations more quickly (not just ambiguous locations) I would delete this finding from the abstract (lines 31-32).

The lack of effect of the grooming condition could very well be linked to general positive bias resulting from positive care (lines 37-39). However, this conclusion would be more convincing if linked to an overall positive bias (as in McGuire et al., 2015) rather than solely a lack of effect of the manipulation. Please reword. It does appear that the goats did exhibit an overall positive bias, based on their responses to the middle location in Figure 3. Did you perform a post-hoc analysis to see which latencies differed? It is important to show whether response to the middle location was the same as to the reward and different from the non-rewarded location. This appears to be a critical omission.

On line 57 please insert an “and” or “or” between depression and anxiety.

On line 62, right off the bat, you identify increased heart rate as associated with agitation in a negative context, but later you make the case for increased heart rate as a sign of positive emotional state. I think this is confusing. Provide a different example here and be clear about the ambivalent findings regarding heart rate later. You later mention that, in cattle, grooming is associated with a decreased hear rate (lines 96-97). The reviewers highlighted the confusing nature of this data as well and little has been done to clarify in the current version of the MS.

Technically, McGuire et al. found no effects of OT on cognitive bias (line 79), although we found the rats to exhibit positive biases in general regardless of condition.

You are missing a period on line 95.

Re-state lines 129-130 in past tense.

Clarify around line 132 the minimal amount of grooming that control goats received.

Please indicate which goats from each condition were trained to expect food on left versus right to show that it was counterbalanced within groups (lines 176-178). Refer to Table 1.

I am confused by the statement on lines 183-185. How do two positive and two negative trials equal 6 trials? Please be clear as to exactly how many trials, of what type, were presented in each session in each phase.

The line 198-199 is unclear. Please reword.

I’m still not clear on the criterion established for passing training. Was the average latency recorded within each session or across all sessions? Presumably not all 19 goats reached that criterion exactly on day 9. Please clarify.

Did you control for odor cues in the unfilled bucket?

A naïve observer should recode the latencies from video for reliability purposes (at least 20% of the trials).

On lines 226-228, please be clear if four goats were naïve and not used in Exp. 1 even if it can be inferred. Please be more explicit throughout.

The habituation period for the BioHarnass appears very shot. Please comment within the text.

Delete the “a” before “software” on line 244.

Should “clear” be “clearly” on line 246?

Experiment 2 needs more of a set-up before the discussion so that the reader is aware of the point of the study. Again, you clarified this in your response letter, and in the discussion, but not in introduction to the study itself.

On line 333, delete the extra s on “years”.

Delete the “the” before “age” on line 337.

Delete the “for” on line 372.

You talk about a faster approach to the ambiguous corridors in younger goats, but did age interact with condition, or was it just an overall effect of age? Please restrict discussion to the results obtained.

Please left-align columns in Table 1.

Why transform the latency data in Figures 2 and 3? This adds to the confusion.

Figure 4 should not depict continuous data. Control and grooming times/conditions are discrete.

---

## Round 0.3 · accepted · Accept

Thank you for your careful attention to the last round of revisions.